# Investigating Clinical Excellence and Impact Awards (INCEA): a qualitative study into how current assessors and other key stakeholders define and score excellence

Bethan M Treadgold [1], John L Campbell,[1] Gary A Abel [1], Jon Sussex,[2] Robert Froud,[3] Lucy Hocking,[2] Emma Pitchforth[1]

¹Primary Care Research Group, University of Exeter Medical School, Exeter, UK
²RAND Europe, Cambridge, UK
³Clinvivo, Edenbridge, UK

**Correspondence to**
Dr Bethan M Treadgold;
b.m.treadgold@exeter.ac.uk

## ABSTRACT

**Objectives** The National Clinical Excellence Awards (NCEAs) in England and Wales were designed, as a form of performance-related pay, to reward high-performing senior doctors and dentists. To inform future scoring of applications and subsequent schemes, we sought to understand how current assessors and other stakeholders would define excellence, differentiate between levels of excellence and ensure unbiased definitions and scoring.

**Design** Semistructured qualitative interview study.

**Participants** 25 key informants were identified from Advisory Committee on Clinical Excellence Awards subcommittees, and relevant professional organisations in England and Wales. Informants were purposively sampled to achieve variety in gender and ethnicity.

**Findings** Participants reported that NCEAs had a role in incentivising doctors to strive for excellence. They were consistent in identifying 'clinical excellence' as involving making an exceptional difference to patients and the National Health Service, and in going over and above the expectations associated with the doctor's job plan. Informants who were assessors reported: encountering challenges with the current scoring scheme when seeking to ensure a fair assessment; recognising tendencies to score more or less leniently; and the potential for conscious or unconscious bias in assessments. Particular groups of doctors, including women, doctors in some specialties and settings, doctors from minority ethnic groups, and doctors who work less than full time, were described as being less likely to self-nominate, lacking support in making applications or lacking motivation to apply on account of a perceived likelihood of not being successful. Practical suggestions were made for improving support and training for applicants and assessors.

**Conclusions** Participants in this qualitative study identified specific concerns in respect of the current approaches adopted in applying for and in assessing NCEAs, pointing to the importance of equity of opportunity to apply, the need for regular training for assessors, and to improved support for applicants and potential applicants.

## STRENGTHS AND LIMITATIONS OF THIS STUDY

⇒ An inclusive sample of key informants were interviewed, which included Advisory Committee on Clinical Excellence Awards (ACCEA) subcommittee assessors (professional, employer and lay assessors) and representatives of professional organisations such as Royal Colleges and those representing particular groups of doctors.

⇒ Interview guides were appropriately and comprehensively informed by the project's initial literature review, the aims of the research and feedback from a dedicated public advisory group.

⇒ Despite initial expressions of willingness to participate in the study, we were unable to recruit many non-white key informants or those from younger age groups.

⇒ We carried out our research alongside a national consultation held by ACCEA which resulted in the move to Clinical Impact Awards and was not known to the research team during the research.

## INTRODUCTION

### Background

Payment systems for doctors can have significant impacts on staff recruitment and retention, quality of care, and costs, and efficiency within a health system.[1] These payment mechanisms vary by employment and financing models of a health system. Since its foundation in 1948, the National Health Service (NHS) in England and Wales has had a performance-related pay system in place to reward senior doctors, dentists and academic general practitioners (primary care physicians with university roles) who make an outstanding contribution to delivering the goals of the NHS. The scheme operates a self-nomination process for the award, and only doctors and dentists are eligible to apply. The ways in which the scheme has been

**Table 1** An overview of the award levels, application criteria and assessment arrangements for the National Clinical Excellence Award (NCEAs) scheme at the time of the research in 2021[4 5 15]

| The clinical excellence awards are awarded on nine local levels and four national levels | | | | | | | | | | | | |
|---|---|---|---|---|---|---|---|---|---|---|---|---|
| Local levels | | | | | | | | | National levels | | | |
| 1 | 2 | 3 | 4 | 5 | 6 | 7 | 8 | 9 | Bronze | Silver | Gold | Platinum |
| £3016–£36 192 | | | | | | | | | £36 192 | £47 582 | £59 477 | £77 320 |

| Applications are assessed following the submission of evidence relating to five domains of performance, which need to be fulfilled in order to be considered for an NCEA | | | | |
|---|---|---|---|---|
| Domain 1: Delivering a high-quality service | Domain 2: Developing a high-quality service | Domain 3: Leading and managing a high-quality service | Domain 4: Research and innovation | Domain 5: Teaching and training |

| Sixteen regional subcommittees of ACCEA in England and Wales, comprising of professional, employer and lay members assess applications for national-level awards. Each domain is independently scored by multiple trained assessors, using a four-point scale | | | |
|---|---|---|---|
| 0: Does not meet contractual requirements or insufficient information produced to make a judgement | 2: Meets contractual requirements | 6: Over and above contractual requirements | 10: Excellent |

ACCEA, Advisory Committee on Clinical Excellence Awards.

maintained has differed across UK nations following devolution,[2 3] but the scheme has been retained in England and Wales where, until early 2022, it was known as the clinical excellence awards (CEAs) scheme. The scheme, with awards offered at a national level, is overseen by an arms-length committee of the UK Department of Health and Social Care, the Advisory Committee on Clinical Excellence Awards (ACCEA).[4] The scheme is designed to reward those who deliver above the standards expected of a consultant, academic general practitioner or dentist fulfilling the requirements of their post.[5] The awards account for a significant amount of public funds. The total value of national awards from the 2020/2021 round was £113 million, with 2046 doctors in receipt of CEAs (1077 bronze awards, 662 silver awards, 221 gold awards, 86 platinum awards), which at the time were pensionable.[5] This award is paid in addition to NHS salary and independent earnings from private practice. See table 1 for an overview of the award levels, application criteria and assessment arrangements for the National Clinical Excellence Award (NCEAs) scheme at the time of the research in 2021.

ACCEA advises health ministers on award allocation following a national competition and assessment process. Despite its prominence, longevity and associated use of public funds, there has been relatively little research into the scheme. Commentators have questioned the scheme from a health economics perspective, including the underlying incentive structure and more recently a study used theoretical frameworks to explore the intrinsic and extrinsic motivations of the medical profession in relation to CEAs.[6–8] Furthermore, concerns have been raised regarding the lack of evidence as to whether the scheme incentivises team performance or benefits patient outcomes, and that inequities lie in the allocation of awards by gender, ethnicity and medical specialty.[6]

In relation to the assessment processes, earlier analysis of historical data showed that the arrangements for assessing applications were defensible, depending on the level of reliability judged to be required in the assessment process.[9] The authors also suggested further research to reform approaches to scoring, and to ensure a scoring system that is sensitive to differences between applicants.

In 2021, ACCEA, launched a national consultation to inform revisions to the awards scheme. A core part of the revised scheme was to adopt a scoring system that is robust, equitable, able to distinguish between levels of excellence, and aligned with ACCEA's overall goals. Our independent study was commissioned to support this ambition. In addition to quantitative analysis of data from applications to ensure assessor reliability, qualitative enquiry allows deeper understanding of the views and experiences of assessors of applications. In informing a revised scoring scheme for the NCEAs, it is important to understand the views and experiences of current arrangements among assessors and other key stakeholders. In 2022, the CEAs have been updated to 'clinical impact awards' although just one round of applications, scoring, and awarding has been completed under these revised arrangements.[10] Our research focuses, therefore, on CEAs with anticipation that the findings are relevant to the revised scheme. We sought to understand how current assessors of NCEAs and other stakeholders would define excellence, implement scoring to differentiate between levels of excellence, and ensure that both the definition of excellence and the assessment and scoring of applications are unbiased.

## METHODS
### Design
Semistructured interviews were conducted with key informants of the NCEAs scheme, between July and August

2021. Reporting of this study is guided by Standards for Reporting Qualitative Research.[11]

## Sample and recruitment strategy

We sought to recruit up to 25 key informants, to include a sample of current ACCEA subcommittee assessors (professional, employer and lay assessors), applicants, representatives of professional organisations and other key stakeholders. We purposively sampled informants on the basis of current membership of ACCEA subcommittees in England and Wales or through membership of relevant national organisations such as Royal Colleges or groups representing doctors. Current ACCEA subcommittee assessors were invited via email invitation from the ACCEA main committee on behalf of the research team. Representatives of national-level organisations and bodies were identified in consultation with the ACCEA main committee and invited via email from the research team. Replies were directly to the research team to prevent the ACCEA main committee from knowing who had replied. From those who expressed interest, we purposively selected a range of individuals to maximise variation in terms of profession, gender and ethnicity. Participant's identity was kept entirely confidential within the research team and not disclosed to any external agencies or bodies, including funding bodies and ACCEA.

## Interview materials and procedure

Two semistructured interview topic guides were developed; one for use with assessors, and one for use with representatives of professional organisations and other stakeholders. The topic guides were informed by a review of literature, the research team's knowledge of the NCEA scheme, regular liaison meetings with the main committee for ACCEA, and input from a project-specific patient and public involvement group. We piloted the topic guides with a previous assessor, known to the research team. Both topic guides were organised into three sections, reflecting the aims of the study: (1) role and experience in relation to assessing and applying for CEAs; (2) 'excellence': what should be rewarded and how should excellence be defined and (3) equity within the application and assessment process. Interviews lasted up to 1 hour and were conducted via online platforms (Microsoft Teams or Zoom) or on the telephone according to interviewee preferences.

## Patient and public involvement

A patient and public advisory group provided written and verbal feedback on the interview topic guides and on initial analysis of the interviews. Resulting changes made to the interview guide included providing a list of current assessment domains and a description of the current scoring scale for key informants.

## Data processing and analysis

Interviews were audio recorded and transcribed. Transcripts were checked by BMT and anonymised. Interview transcripts were analysed inductively with a reflexive thematic analysis consisting of six recursive phases.[12 13] Together, BMT and EP developed an initial list of subthemes. Individual codes with similar semantic meanings were grouped together into subthemes, and stand-alone codes were transferred directly into subthemes. Potential overarching themes were developed and iteratively refined with JLC. JLC brought relevant experience to inform the overall study. To facilitate reflexivity, JLC was not involved in data collection and only in late stages of analysis and interpretation.

In a reflexive thematic analysis, the coding process is unstructured, which meant that codes evolved to capture the researchers' deepening understanding of the data and research area. During the six phases of analysis, BMT reflected on her assumptions surrounding the data and research area and how they might shape analysis. Professional characteristics of the core research team were also reflected on during the final analysis and write up stages, to ensure that such characteristics did not inappropriately shape findings. The results presented in this paper focus on the findings relating to how clinical excellence is defined and considerations of equity in scoring. Detailed comments made around the numerical scoring scales informed subsequent stages of research.

## Findings

### Characteristics of key informants

Twenty-five key informants were interviewed. Most were ACCEA subcommittee assessors (n=14), others held dual roles as assessor and representative of professional organisation (n=6) or were solely representative of professional organisation (n=5). Six key informants reported holding an NCEA, and one participant reported being unsuccessful in application. A near equal proportion of interviewees were female (n=12) and male (n=13). Despite concerted efforts to recruit a diverse ethnic sample through purposive sampling, most key informants were of white ethnic background (n=19). A small number of informants were of other ethnic backgrounds (Indian or Indian Mmixed background n=3, African or African mixed background n=1, Chinese or Chinese mixed background n=1). Nearly all informants were aged 45 years or older (n=24). Eleven potential key informants, who had expressed an interest in being involved in the interviews, did not respond to further invitations to participate.

### Overarching themes

Our inductive reflexive thematic analysis generated three overarching themes around how interviewees defined clinical excellence, differentiated between levels of excellence and discussed definitions for and scoring of excellence that are unbiased. The overarching themes and their subthemes explore how: (1) 'Clinical excellence' is multifaceted, and a range of behaviours and activities should be rewarded; (2) Assessors develop their own personal strategies to guard against bias and perceived challenges with the current scoring scheme and (3) There are perceived inequities for marginalised groups of

**Table 2** Illustration of overarching and descriptive themes from qualitative interviews around defining clinical excellence, scoring and equity

| How do current assessors of NCEAs and other stakeholders define excellence, implement scoring to differentiate between levels of excellence, and ensure that definitions of excellence and scoring are unbiased? | |
| --- | --- |
| 'Clinical excellence' is multifaceted, and a range of behaviours and activities should be rewarded | Demonstrating going over and above job expectations |
| | Making an exceptional difference to patients and the National Health Service |
| | Recognising the context or setting through which excellence is achieved |
| | Outlining the impact of excellence |
| Assessors develop their own personal strategies to guard against bias and perceived challenges with the scoring scheme | Obtaining examples of relevant job plans to tackle the diversity of specialties |
| | Modifying scores to reflect application structure and content |
| | Maintaining a 'hawk' or 'dove' assessment style to address unconscious bias |
| | Considering various sources of excellence evidence presented |
| There are perceived inequities for marginalised groups of doctors in producing evidence and due to the self-nomination process | Difficulties for certain specialties to generate suitable evidence |
| | Gender and ethnic disadvantages incurred through the self-nomination process |
| | Challenges for less than full-time applicants to generate evidence |
| | Widespread penalisation for teamwork |

NCEAs, National Clinical Excellence Awards.

doctors surrounding producing evidence and due to the self-nomination process. A number of subthemes were also identified (table 2).

## 'Clinical excellence' is multifaceted, and a range of behaviours and activities should be rewarded

Informants' views on what constitutes clinical excellence appeared to be related to their current role as a medical professional, and/or past experiences as patients of the NHS.

Nearly all informants reported that clinical excellence means demonstrating going over and above job expectations and the expectations of their colleagues, in terms of hours worked, taking on additional and more senior tasks, and relieving workload of colleagues:

> …to be able to really demonstrate what do I do that's over and above. My job plan says this, my contract says that, actually I have excelled in the way that I have delivered on my contract … (Female; White ethnicity; Lay assessor).

Some also argued that doctors should not be further rewarded for activities that they are already being remunerated for:

> A lot of these posts such as clinical director or divisional director are renumerated at quite a high level so divisional director is getting paid 30 thousand pounds in our hospital on top of their NHS salary. So I don't feel that… should carry weight in terms of clinical excellence award because they're already being paid for that… (Male; White ethnicity, Professional assessor, and award holder).

Many informants explained that making an exceptional difference to patients and the NHS constitutes clinical excellence and should be rewarded, being a perceived fundamental aim of the NHS and those working within it. A marker of excellence included patient preferences to be treated by certain doctors:

> 'I want to go and see this doctor for this disease,' because the care that they give or the team that delivers that care is just brilliant and they're fabulous. We want to reward that—that's clinical excellence. (Male; White ethnicity; Professional assessor, Representative of professional organisation, award holder).

Many informants explained the importance of outlining the impact of excellence, such as on patients, their colleagues and the NHS overall. Some informants noted that as this is an NCEA Scheme, national and international impact of activities or otherwise beyond local impact should be demonstrated and rewarded:

> I think clinical excellence should be where people are… really showing that excellence at the regional or national level. (Female; non-White ethnicity; Professional assessor).

But others noted that excellence demonstrated at local level should not be overlooked:

> And if you read the definitions of national it doesn't have to be national it can be anything outside of

your immediate Trust, so you could be applying for a national award because you've influenced the Trust next door. (Female; White ethnicity; Professional assessor).

In assessing clinical excellence, many informants described the importance of recognising the context or setting through which excellence is achieved. Challenges for applicants were reported in respect of generating research excellence in non-university hospitals, or within specialties that offered less time and fewer resources to undertake research, such as emergency medicine and radiology:

> … the man or woman sitting in District Hospital X where they're struggling just to manage the volume of work, and they cannot score particularly highly in domain four (research and innovation) because they just don't have the time, whereas the individual who's got half their week as an academic can get big publications. (Male; non-White ethnicity; Professional assessor and past award holder).

### Assessors develop their own personal strategies to guard against bias and against perceived challenges with the scoring scheme

Key informants who were assessors explained how they had developed personal strategies to assess clinical excellence and to try to ensure fairness. Strategies appeared to be in response to assessors' experiences, and the written applications themselves. These informants did not specify that such strategies were recommended or guided by ACCEA, but were rather strategies that they had personally developed.

Assessors described obtaining examples of relevant job plans to tackle the diversity of specialties and job roles in applications that were assigned to them. Lay assessors, and occasionally other assessors, suggested that their lack of knowledge about the typical working day of some specialties, especially with sometimes limited details provided under the job plan section, made it difficult for them to assess what constitutes a contribution that was 'over and above'. They, therefore, sought information about the job role in focus:

> I know a little bit because we've got a big neurosciences centre… But I'm merely a lay person and I'm judging someone's form on neuroscience, because what do I know? So, if the Societies give me a leg-up and support me that helps. (Male; non-White ethnicity; Professional assessor and past award holder).

In assessing awards, informants reflected on the subjectivity of the process. Informants who were assessors described maintaining a 'hawk' or 'dove' assessment style to address unconscious bias, and that being aware of one's own potential biases and performance as an assessor was important to diffuse issues with subjectivity:

> If you were a hawk you need to be a consistent hawk. You can't be a hawk for one person and a dove for

another… (Female; non-White ethnicity; Professional Assessor).

Many noted several factors that could shape one's perception and expectation of excellence, such as the assessors' own job, life experiences, protected characteristics and whether the applicant is a current award holder.:

> I do four days of [speciality], 7am to 7pm… So my standard is really very high because of what I do for a living… (Male; White British ethnicity; Professional Assessor, representative of professional organisation, and award holder).

Assessor-informants described modifying scores to reflect application structure and content, when applicants' presentation of material was particularly difficult to read. Examples included the use of undefined acronyms, variable use of bulleting and the use of excessive and poorly focused sentences and presenting the same evidence in multiple domains or presenting evidence in the incorrect domains:

> … abbreviations …Sometimes they bother to explain them earlier and sometimes they don't…If you're marking them and it's all bullet-points, it's pretty easy to scan through, but often you don't get a sense of a journey or a development. Whereas if it's all dense text, it's much more difficult to read through…. (Male; White ethnicity; Professional assessor, representative of professional organisation, and award holder).

Informants also emphasised the importance of considering several sources of excellence evidence presented, and that assessors need a comprehensive overview of applicants' excellence, in terms of role and contribution, from the applicants' emic perspective, as well as the etic perspective of colleagues and patients, in order to fairly score:

> All evidence submitted says this person is a wonderful person you know they're really good at their work… so yes if it was obligatory that we had to have citations from a number of different people then that would make it better. (Female; White ethnicity; Professional Assessor).

Some informants suggested that citations from Royal Colleges were not entirely helpful and sometimes overlooked, as they were generally positive, and not all applicants were a member of a College.

### There are perceived inequities for marginalised groups of doctors in producing evidence and due to the self-nomination process

Key informants discussed a variety of inequities within the assessment scheme, which were suggested to be underpinned by long-standing inequities in the NHS. Informants explained that such inequities act as a perceived barrier to being successful in the scheme and thus discourage doctors from applying.

Informants from various clinical specialties reported that there are difficulties for certain specialties to generate suitable evidence, such as in pathology, anaesthesia and emergency medicine, where doctors may not be able to generate appropriate evidence for all domains as easily as doctors from other specialties:

> Radiologists sit in a dark room looking at pictures and don't have that much contact with many patients—so how do you assess how their patient benefit is, and how they get quality feedback from a patient who may never have seen them? Anaesthetists have a similar sort of problem… (Male; White ethnicity; Professional Assessor, representative of professional organisation, and award holder).

Other informants suggested that it is difficult for service-focussed doctors, especially those working in district general hospital settings, to generate as much research evidence as academic-focussed doctors working in university teaching hospitals. Some informants also explained that there is a general assumption among doctors that most of these awards are given to doctors working within some specialties, such as surgery; potentially dissuading doctors from other specialties from applying.

Many informants also reported time challenges for less than full-time applicants to generate evidence, because of the perception that they are unable to go over and above their work hours, or to take on additional tasks, and thus have restricted opportunities to develop evidence across all five domains, unlike full-time applicants, who were suggested to be able to do this:

> …because you've only got a 0.8 whole time equivalent contract, and so what you're delivering at your baseline appears to be lower than what someone who's on a full-time contract delivers, and then you've got to then not only bridge that gap, but also then go over and above, and I think that's really difficult to demonstrate. (Male; White ethnicity; Professional assessor, representative of professional organisation, and award holder).

Eight informants explained that there are gender and ethnic disadvantages incurred through the self-nomination process of the scheme. Male and female informants, and informants from a variety of ethnic backgrounds, explained that female doctors and doctors from ethnic minority groups are less likely to recognise their excellence and to self-nominate themselves for an award; and when they do so, their presentation of material is sometimes seen as too modest:

> I think another one from somebody who is not British, for example, and there is a very different way of writing, there's a very different way of expressing themselves, sometimes a lot more modest. You look at the difference between a female applicant and a male applicant and how they present their evidence. (Female; White ethnicity; Lay assessor).

It was suggested that female doctors and doctors from ethnic minority groups do not always feel comfortable taking credit for team effort in a self-nomination process application:

> As an ethnic minority person, you do suffer from imposter syndrome unlike say my white colleague who might have done half the things that I did but are absolutely fine in going there. Secondly, if you're a female, then it's even worse because you never really reach that stage where you think that you're good enough until somebody tells you. (Female; non-White ethnicity; Professional assessor).

Informants from a range of specialties and backgrounds explained that there is widespread penalisation for teamwork in the application. Informants explained that the narrative nature of the application focused on the applicant only, and that mentioning too much teamwork could be costly:

> You can spend a lot of time writing things: 'I did this, I did that.' Or was it, 'We did this or we did that, or was I part of the team that did this?' and what's more powerful. I think that plenty of folks are less good at taking the credit for stuff than others… (Male; White ethnicity; Professional assessor, representative of professional organisation, and award holder).

Nearly all informants provided practical suggestions which might be considered in relation to improving the scoring process and equitability of the scheme. Many suggested the need for advertising the scheme more broadly, such as through members of ACCEA visiting trusts to talk about the scheme to underrepresented specialties and groups of doctors, and allocating champions within trusts to encourage applications. Many also recommended that ACCEA should create support materials for prospective applicants, such as videos about how to collect evidence, or whom to approach for advice on applications. Finally, some informants advised that for assessors; more frequent training, contact and information about the range of specialties for assessors, would improve their contribution.

## DISCUSSION
### Summary of findings
Our qualitative study suggests that assessors and members of professional organisations believe that excellence must constitute going 'over and above' the expectations of job plans, be related to service to the NHS, and demonstrate benefits to patients. The interviews further highlighted that assessing whether an applicant has demonstrated such excellence in their clinical practice could be challenging, given that often limited job plan information is provided to assessors. Assessors in our study emphasised the subjective nature of assessment and explained that they may develop their own personal strategies to guard against their own potential biases, and against challenges

encountered with assessing the evidence presented. Such strategies did not necessarily correspond with ACCEA assessor guidelines.[14] Challenges included the diversity of job roles represented in applications and variation in the presentational quality of evidence submitted. Informants also described potential inequities within the assessment scheme which may disadvantage some groups of doctors. Female doctors, doctors in some specialties and settings, doctors from minority ethnic groups, and doctors who work less than full time were considered to be less likely to self-nominate, to lack support or to be deterred by a perceived likelihood of not being successful.

## Findings in context of existing research

Concerns have previously been highlighted regarding potential inequities in the CEA scheme relating to disparities in the number and success of applications by ethnicity, gender and clinical specialty of applicant.[4] Our findings identify an awareness that the subjective nature of scoring may contribute to inequalities, for example, that assessors develop their own strategies to overcome unconscious bias. However, assessors perceive that fairness in scoring is only one element of ensuring equity and the nature of the scheme, particularly in the need for self-nomination, may mean that some groups are disadvantaged more than others. Female doctors, and doctors from ethnic minority groups, were considered by informants to be less likely to recognise excellence in their performance, to take credit for team effort, and to nominate themselves for an award. Informants suggest that when they do apply, their presentation of material can be too modest, undermining their chances of success. Our findings are consistent with previous analysis that shows that some specialties, such as psychiatry and anaesthetics, are under-represented among CEA applicants.[6] Challenges were perceived for doctors in these specialties in terms of generating evidence around delivering high-quality service and obtaining feedback. It may be that key informants underplay the role of assessors in contributing to any disparities, but the findings have important implications for the ways in which the scheme is advertised, and potential applicants are identified and supported. Improved guidance for scoring may also reduce the need for assessors to develop their own strategies.

## Strengths and limitations

To our knowledge, this is the first qualitative exploration of the views and experiences of current assessors and representatives of professional organisations, about the process of assessing applications for NCEAs. We have examined how the NCEAs are viewed in terms of the meaning of clinical excellence and the behaviours that should be rewarded, along with the processes through which assessors assess applications and ensure fairness, as well as the potential negative influences of specialty, gender, ethnicity and part-time working on applying. We achieved a good gender and professional role balance

among informants to ensure that a range of views and experiences were captured.

Despite initial willingness to participate in the study, we struggled to recruit many non-white key informants, as well as doctors from younger age groups. Furthermore, during the recruitment process we did not collect information on some other protected characteristics such as disability, and nor did we capture the views of representatives of employer organisations. It is, therefore, possible that we omitted to capture some extant definitions of clinical excellence, experiences with scoring and reflections on inequities important to some groups, organisations and individuals. In addition, it should be noted that the majority of key informants had experience of scoring as subcommittee members and were likely to be invested to some extent in the NCEA Scheme. The language used in interviews such as 'over and above' and 'outstanding' reflected some of the languages used in ACCEA policy and guidelines.[15] Views of a broader range of stakeholders may have brought additional insights to understanding how excellence may be defined. That said, with a focus on scoring, the key informants were well placed to speak from experience, and our findings show that despite guidance on scoring and, despite being part of the system, they were willing to describe personal strategies that they employ to overcome some of the perceived limitations of the scheme. Further work might focus on triangulation or corroborating of these findings with empirical work to establish the extent to which such views are held in the population from which we purposively sampled.

## Implications and future research

In consultation with ACCEA, these findings are being used to inform the development of a revised scheme, anticipated to run from 2023 to 2024, which will aim to be more understandable and practical for assessors and applicants, and will provide clear guidance on carrying out a valid, an equitable assessment of applications. The practical suggestions from informants around advertising the scheme, supporting applicants and supporting assessors are also being explored by ACCEA.

Future research could involve further qualitative interviews to explore the views and experiences of assessors and applicants with the revised NCEAs scheme around the practicalities, and equitableness of the scoring process, as well as the fairness of providing evidence for the application. Further interviews could also explore whether new training materials developed by ACCEA to support prospective applicants to apply have, as intended, reduced potential barriers for marginalised groups of doctors.[14] Finally, future research should explore the role of patient outcomes in the assessment of NCEA applications, given existing concerns around the general lack of evidence on patient outcomes in current assessment.[4]

## CONCLUSIONS

The NCEAs represent a significant opportunity for senior doctors and dentists in England and Wales and use a considerable amount of taxpayers' money. Our study suggests that current assessors and representatives of professional organisations are supportive of the NCEAs in being able to reward clinical excellence. The findings are, at the time of writing (Spring 2023), in the process of informing future scoring and assessment of applications. Our findings also point to the broader importance of equity of opportunity to apply, and the importance of regular training for scorers and support for applicants.

**Contributors** BMT designed the qualitative interview guide, recruited key informants, carried out qualitative interviews, qualitatively analysed the data and prepared the manuscript. JLC, the guarantor of the study, prepared the grant application/protocol, helped design the qualitative interview guide, was involved in the qualitative analysis and contributed to the written manuscript. GAA and JS prepared the grant application/protocol, helped design the qualitative interview guide and contributed to the written manuscript. RF prepared the grant application/protocol, contributed to the design of the qualitative interview guide and contributed to the written manuscript. LH contributed to the design of the qualitative interview guide and contributed to the written manuscript. EP prepared the grant application/protocol, designed the qualitative interview guide, helped to recruit key informants, qualitatively analysed the data and contributed to the written manuscript.

**Funding** This research was commissioned by the Department of Health and Social Care through the National Institute for Health Research Policy Research Programme (project reference NIHR202992).

**Competing interests** BMT, GAA, JS, RF, LH and EP declare no completing interests. JLC holds a national clinical excellence award. JLC has previously been both a member and Chair of a Regional Subcommittee of ACCEA, although had no involvement with ACCEA administration at the time the research was commissioned and undertaken.

**Patient and public involvement** Patients and/or the public were involved in the design, or conduct, or reporting, or dissemination plans of this research. Refer to the Methods section for further details.

**Patient consent for publication** Consent obtained directly from patient(s).

**Ethics approval** Key informants were invited to participate in the research based on their professional role. Formal ethical approval was not sought based on screening the study outline using the UKRI/NHS Health Research Authority online assessment tool. Key informants were provided with information about the study and written consent was obtained before participation in the interviews. Informants were given full information about handling and storage of resulting data. Participation was voluntary, and data were securely and anonymously stored. In reporting the findings of this study, we provide brief labels for quotations about the characteristics of key informants (eg, about the ethnicity of the informant), to limit potential identification of informants.

**Provenance and peer review** Not commissioned; externally peer reviewed.

**Data availability statement** No data are available.

**ORCID iDs**
Bethan M Treadgold http://orcid.org/0000-0002-0255-7422
Gary A Abel http://orcid.org/0000-0003-2231-5161

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
