## [Reviewer comments · BMJ Open]

ARTICLE DETAILS

TITLE (PROVISIONAL)	Investigating Clinical Excellence and Impact Awards (INCEA): A qualitative study into how current assessors and other key stakeholders define and score excellence
AUTHORS	Treadgold, Bethan; Campbell, John; Abel, Gary; Sussex, Jon; Froud, Robert; Hocking, Lucy; Pitchforth, Emma

VERSION 1 – REVIEW

REVIEWER	Mark Exworthy University of Birmingham, HSMC
REVIEW RETURNED	06-Dec-2022

GENERAL COMMENTS	This paper presents research on the views of the assessment process relating to Clinical Excellence Awards; it is topic which would be suitable for BMJ Open. The paper is structured well. The methods are described in detail. The conclusions are a logical consequence of the preceding arguments. Whilst there is much to admire about the paper, there are several areas which lack sufficient detail and are uncritical. These would need to be addressed before publication. The authors need to clarify the scope of CEAs in three main ways. First, international readers (and probably some British ones too) need a clearer description and explanation of the peculiar remit of CEAs; for example, CEAs are a form of ‘performance-related pay’ (in addition to NHS salary and independent of earnings from private practice), and that only doctors are eligible. These are not included in the abstract, for example. Second, the authors need to acknowledge the (relatively recent) separation of local and national CEAs across what were 13 categories (level 1-9 and then Bronze to Platinum). The authors allude to “regional and national” awards in one quote (p.8, line 45) but this is insufficient. Indeed, the distinction within the national CEAs needs to be explained; this might involve discussion about progression from ‘lower’ to ‘higher’ awards. The findings do not mention the award levels (bronze, silver etc) and so do not discuss how ‘excellence’ between, say, a Bronze and Silver award might be judged by assessors. Similarly, whilst the domains are mentioned (p.4), the scoring of them is less well elaborated. The scoring scales are not mentioned and yet, the authors state that “Detailed comments made around the numerical scoring scales informed subsequent stages of research “ (p.6, line 44). The authors do note that doctors in some specialties may have more opportunities to demonstrate ‘excellence’ than others, and yet the implications of differential scores across these domains are not discussed. Hypothetically, a mix of high and low scores in the domains might be equivalent to even scores across all of them, but
---

	these two cases would have different meanings of `excellence.` Third, the wider system of awards needs to be acknowledged, for example, in terms of costs. CEAs awards (numbers or costs) in previous rounds are not acknowledged, nor are previous pension liabilities. The authors claim (as there is not citation) that the `total` cost of the 532 national awards in 2022 is £129 million. This would mean that the average award value of £242,481 (129m/532) which, without adequate explanation, would seem incredible. The authors have made various assumptions that need to be addressed. One can understand why the authors take the CEA scheme as a given. Although the terms such as `over and above`, `exceptional` and of course, `excellence` are used in policy documentation, one might expect the authors to be questioning. The subjective nature of CEA scoring is stated (eg. p.9, line 32; p.12, line 3) but the consequences need to be fully explored. For example, it is unclear whether the actions (table 2 / p.5) are ACCEA guidance or individual (assessor) strategies; there are implications for uneven uptake if the former and for good governance if the latter. The authors indicated these as “personal strategies (p.9, line 13) and also pointed to individual insights (“I know a little bit” p.9, line 26). Also, the authors could have acknowledged better that the interviewees would be expected to be sympathetic and/or supportive of CEAs; this point needs to be noted in interpreting the findings. For example, the quotes did suggest some post-hoc rationalisation whereby interviewees sought to self-justify their decisions to the researcher (eg. p.9, line 45). The lack of corroboration or triangulation (such as different sources of evidence and/or validation of findings) dents the credibility of the findings. Also, as one of the co-authors holds a national CEA and was involved in the study throughout (as stated in the authors’ contributions; p.14), it is unclear how any bias was handled. The findings needed to highlight better the distinctions between the categories of the interview sample. The context for each person quoted was helpfully given but the analysis did not draw any distinctions. For example, how did the view of the six interviewees who held a national CEA differ, if at all, from the others. Also, the sample of 25 did not include a representative of employer organisations (p.12, line 54); this is a significant omission. One might expect significant differences between categories such as professional organisation and employer. Indeed, the incorporation of employer/managerial and lay perspectives some twenty years ago was designed to bring alternative viewpoints into the decision-making process. Finally, , while the opening section (p.4, line 53) noted wider concerns about the general lack of evidence on patient outcomes in CEA applications, this was not apparent in the findings – did interviewees not discuss it? If so, that would be significant. In summary, I recommend that the paper is revised and resubmitted.
--	--

REVIEWER	Ishanka Talagala Ministry of Health Nutrition and Indigenous Medicine, National programme for prevention and control of NCDs
REVIEW RETURNED	10-Dec-2022

GENERAL COMMENTS	Congratulations to the authors on this important study. Please see the following suggestions: Page 7, line 21: “... given full information about what handling...”
--

	here, 'what' should be removed. Page 14, line 6: “..anticipated to run from 2022/2023..” Need to change the years for at least 2023/2024, as we are now at the end of the year 2022, and it will take some time for the ACCEA to develop a novel system incorporating the findings of the study.
--	--

VERSION 1 – AUTHOR RESPONSE

Reviewer: 1 Prof. Mark Exworthy, University of Birmingham	
Reviewer comment	Response to reviewer
This paper presents research on the views of the assessment process relating to Clinical Excellence Awards; it is topic which would be suitable for BMJ Open.	Thank you for supporting the suitability of this paper for BMJ Open.
The paper is structured well. The methods are described in detail. The conclusions are a logical consequence of the preceding arguments.	Thank you for highlighting the clear structure of the paper.
Whilst there is much to admire about the paper, there are several areas which lack sufficient detail and are uncritical. These would need to be addressed before publication.	We have endeavoured to address each comment.
The authors need to clarify the scope of CEAs in three main ways.	We have now clarified the scope of the CEAs in each of the three main ways, as described below.
First, international readers (and probably some British ones too) need a clearer description and explanation of the peculiar remit of CEAs; for example, CEAs are a form of 'performance-related pay' (in addition to NHS salary and independent of earnings from private practice), and that only doctors are eligible. These are not included in the abstract, for example.	Thank you for the suggestion. In the background section of this paper, we have included these additional details. The opening paragraph of the background section now reads (page 4 marked copy): “Payment systems for doctors can have significant impacts on staff recruitment and retention, quality of care, and costs, and efficiency within a health system¹. These payment mechanisms vary by employment and financing models of a health system. Since its foundation in 1948, the NHS in England and Wales has had a performance-related pay system in place to reward senior doctors, dentists and academic general practitioners (primary care physicians with university roles) who make an outstanding contribution to delivering the goals of the NHS. The scheme operates a self-nomination process for the award, and only doctors and dentists are eligible to apply. The ways in which the scheme has been maintained has differed across UK nations following devolution²⁻³, but the scheme has been retained in England and Wales where, until early 2022, it was known as the Clinical Excellence Awards (CEAs) scheme. The scheme, with awards offered at a national level, is overseen by an arms-length committee of the UK Department of Health and Social Care, the Advisory Committee on Clinical Excellence Awards

(ACCEA)⁴. The scheme is designed to reward those who deliver above the standards expected of a consultant, academic general practitioner, or dentist fulfilling the requirements of their post⁵. The awards account for a significant amount of public funds. The total value of national awards from the 2020/21 round was £113million, with 2,046 doctors in receipt of CEAs, (1,077 bronze awards; 662 silver awards; 221 gold awards; 86 platinum awards), which at the time were pensionable⁵. **This award is paid in addition to NHS salary and independent earnings from private practice.** See Table 1 for an overview of the award levels, application criteria, and assessment arrangements for the National Clinical Excellence Award (NCEAs) scheme at the time of the research in 2021.”

The opening section to the abstract now reads (page 2 marked copy):

“Objectives The National Clinical Excellence Awards (NCEAs) in England and Wales were designed, **as a form of performance-related pay**, to reward high performing senior doctors and dentists. To inform future scoring of applications and subsequent schemes, we sought to understand how current assessors and other stakeholders would define excellence, differentiate between levels of excellence, and ensure unbiased definitions and scoring.”

Second, the authors need to acknowledge the (relatively recent) separation of local and national CEAs across what were 13 categories (level 1-9 and then Bronze to Platinum). The authors allude to “regional and national” awards in one quote (p.8, line 45) but this is insufficient. Indeed, the distinction within the national CEAs needs to be explained; this might involve discussion about progression from ‘lower’ to ‘higher’ awards. The findings do not mention the award levels (bronze, silver etc) and so do not discuss how ‘excellence’ between, say, a Bronze and Silver award might be judged by assessors. Similarly, whilst the domains are mentioned (p.4), the scoring of them is less well elaborated. The scoring scales are not mentioned and yet, the authors state that “Detailed comments made around the numerical scoring scales informed subsequent stages of

Thank you for your suggestion to expand these details. We have now included a new table in the background section of the report, which provides an overview of the award levels, application criteria, and assessment arrangements for the National Clinical Excellence Awards scheme at the time of the research in 2021. Table 1 of the background section reads (page 5 marked copy):

The Clinical Excellence Awards are awarded on nine local levels and four national levels

Local levels						National levels		
1	2	3	4	5	6	Bronze	Silver	Gold
7	8	9				Platinum		
£3,016 - £36,192						£36,192	£47,582	
						£59,477	£77,320	

Applications are assessed following the submission of evidence relating to five domains of performance, which need to be fulfilled in order to be considered for a National Clinical Excellence Award

research “ (p.6, line 44). The authors do note that doctors in some specialties may have more opportunities to demonstrate ‘excellence’ than others, and yet the implications of differential scores across these domains are not discussed. Hypothetically, a mix of high and low scores in the domains might be equivalent to even scores across all of them, but these two cases would have different meanings of ‘excellence.’	Domain 1: Delivering a high-quality service	Domain 2: Developing a high-quality service	Domain 3: Leading and managing a high-quality service	Domain 4: Research and innovation	Domain 5: Teaching and training
Sixteen regional sub-committees of ACCEA in England and Wales, comprising of professional, employer and lay members assess applications for national-level awards. Each domain is independently scored by multiple trained assessors, using a four-point scale					
0: Does not meet contractual requirements or insufficient information produced to make a judgement		2: Meets contractual requirements	6: Over and above contractual requirements	10: Excellent	
Table 1. An overview of the award levels, application criteria, and assessment arrangements for the National Clinical Excellence Award (NCEAs) scheme at the time of the research in 2021.⁴⁻⁶					
Third, the wider system of awards needs to be acknowledged, for example, in terms of costs. CEAs awards (numbers or costs) in previous rounds are not acknowledged, nor are previous pension liabilities. The authors claim (as there is not citation) that the ‘total’ cost of the 532 national awards in 2022 is £129 million. This would mean that the average award value of £242,481 (129m/532) which, without adequate explanation, would seem incredible.	We agree. The relevant section of the opening paragraph of the background section now reads (page 4 marked copy): The scheme is designed to reward those who deliver above the standards expected of a consultant, academic general practitioner, or dentist fulfilling the requirements of their post⁵. The awards account for a significant amount of public funds. The total value of national awards from the 2020/21 round was £113million, with 2,046 doctors in receipt of CEAs, (1,077 bronze awards; 662 silver awards; 221 gold awards; 86 platinum awards), which at the time were pensionable⁵. This award is paid in addition to NHS salary and independent earnings from private practice. See Table 1 for an overview of the award levels, application criteria, and assessment arrangements for the National Clinical Excellence Award (NCEAs) scheme at the time of the research in 2021.				
The authors have made various	On reflection we agree, and we have endeavoured to address				

assumptions that need to be addressed.	these in line with suggestions.
One can understand why the authors take the CEA scheme as a given. Although the terms such as 'over and above', 'exceptional' and of course, 'excellence' are used in policy documentation, one might expect the authors to be questioning.	Thank you for this suggestion. The limitations section of the discussion now reads (page 14 marked copy): “Despite initial willingness to participate in the study, we struggled to recruit many non-white key informants, as well as doctors from younger age groups. Furthermore, during the recruitment process we did not collect information on some other protected characteristics such as disability, and nor did we capture the views of representatives of employer organisations. It is therefore possible that we omitted to capture some extant definitions of clinical excellence, experiences with scoring, and reflections on inequities important to some groups, organisations, and individuals. In addition, it should be noted that the majority of key informants had experience of scoring as sub-committee members and were likely to be invested to some extent in the NCEA Scheme. The language used in interviews such as ‘over and above’ and ‘outstanding’ reflected some of the languages used in ACCEA policy and guidelines¹³. Views of a broader range of stakeholders may have brought additional insights to understanding how excellence may be defined. That said, with a focus on scoring, the key informants were well placed to speak from experience, and our findings show that despite guidance on scoring and, despite being part of the system, they were willing to describe personal strategies that they employ to overcome some of the perceived limitations of the scheme.”
The subjective nature of CEA scoring is stated (eg. p.9, line 32; p.12, line 3) but the consequences need to be fully explored. For example, it is unclear whether the actions (table 2 / p.5) are ACCEA guidance or individual (assessor) strategies; there are implications for uneven uptake if the former and for good governance if the latter. The authors indicated these as “personal strategies (p.9, line 13) and also pointed to individual insights (“I know a little bit” p.9, line 26).	Thank you for encouraging clarification on this subjective matter. The title of theme two now reads (pages 8 and 10 marked copy): “Assessors develop their own personal strategies to guard against bias and perceived challenges with the current scoring scheme.” The description of theme two now reads (page 10 marked copy): “Key informants who were assessors explained how they had developed personal strategies to assess clinical excellence and to try to ensure fairness. Strategies appeared to be in response to assessors’ experiences, and the written applications themselves. These informants did not specify that such strategies were recommended or guided by ACCEA, but were rather strategies that they had personally developed.” The relevant section in the summary of findings in the discussion now reads (page 13 marked copy): “Assessors in our study emphasised the subjective nature of

	assessment, and explained that they may develop their own personal strategies to guard against their own potential biases, and against challenges encountered with assessing the evidence presented. Such strategies did not necessarily correspond with ACCEA assessor guidelines¹⁵.
Also, the authors could have acknowledged better that the interviewees would be expected to be sympathetic and/or supportive of CEAs; this point needs to be noted in interpreting the findings. For example, the quotes did suggest some post-hoc rationalisation whereby interviewees sought to self-justify their decisions to the researcher (eg. p.9, line 45).	We agree that the majority of informants were somewhat invested in the scheme, which has some limitations, and we have now amended the limitations section of the discussion (page 14 marked copy): Though it must be noted that while we might expect the involved key informant assessors to be sympathetic and/or supportive of the CEAs, this was not necessarily true of the informants from professional organisations. Indeed, many of the professional informants expressed dissatisfaction in terms of equity within the application process, as described in theme three of our findings ‘There are perceived inequities for marginalised groups of doctors surrounding producing evidence and due to the self-nomination process’. We also note the relative strengths of this group having experience of scoring and being able to reflect on definitions of excellence.
The lack of corroboration or triangulation (such as different sources of evidence and/or validation of findings) dents the credibility of the findings.	We have reflected on your points to limitations of the study and have redrafted the limitations section. We have noted the lack of corroboration and triangulation as an additional limitation, and also emphasised that our qualitative research was to explore what a group of respondents thought, rather than to make generalisations of sampled findings to a population. We trust that our novel findings and outlines for further research outweigh the limitation on balance. The end of the limitations section of the discussion now reads (page 14 marked copy): “Further work might focus on triangulation or corroborating of these findings with empirical work to establish the extent to which such views are held in the population from which we purposively sampled.”
Also, as one of the co-authors holds a national CEA and was involved in the study throughout (as stated in the authors’ contributions; p.14), it is unclear how any bias was handled.	Thank you for recommending that we reflect on this element of the study. The data collection and analysis was led by BT and EP (non-medical health services researchers). JC, who at the time of the study held an NCEA, brought relevant experience to inform the overall study. To facilitate reflexivity, JC was not involved in data collection and only in late stages of analysis and. To this end we have clarified our data processing and analysis section of the methods, which now reads (page 7 marked copy): “Interviews were audio recorded and transcribed. Transcripts were checked by BT, and anonymised. Interview transcripts were analysed inductively with a reflexive thematic analysis consisting of six recursive phases¹⁰⁻¹¹. Together, BT and EP developed an initial list of sub-themes. Individual codes with similar semantic meanings were grouped together into sub-

	themes, and stand-alone codes were transferred directly into sub-themes. Potential overarching themes were developed and iteratively refined with JC. JC brought relevant experience to inform the overall study. To facilitate reflexivity, JC was not involved in data collection and only in late stages of analysis and interpretation. We have additionally updated the competing interest statement (page 15 marked copy): “BT, GA, JS, RF, LH, EP declare no completing interests. JC holds a national clinical excellence award. JC has previously been both a member and Chair of a Regional Subcommittee of ACCEA, although had no involvement with ACCEA administration at the time the research was commissioned and undertaken.”
The findings needed to highlight better the distinctions between the categories of the interview sample. The context for each person quoted was helpfully given but the analysis did not draw any distinctions. For example, how did the view of the six interviewees who held a national CEA differ, if at all, from the others.	Thank you for your comment. The aim of the thematic analysis was to explore common themes around defining excellence, scoring between levels of excellence, and ensuring non-bias, rather than focusing on the context and circumstances around the individual key informants. We included descriptions of the informants alongside each quote (e.g., page 8 “Female; White ethnicity; Lay assessor”), and in various occasions described when certain groups of informants reported holding a common view (e.g., page 10 “Key informants who were assessors explained how they had developed...”, page 11 “Informants from various clinical specialities reported that there are difficulties...”, page 12 “Male and female informants, and informants from a variety of ethnic backgrounds, explained that female doctors and doctors from ethnic minority groups are less likely to recognise...”), though the aim was not to explore deeply into the backgrounds of the informants. We also sought to highlight contrary views in the reporting of themes (e.g., page 9 “But others noted that excellence demonstrated at local level should not be overlooked...”). Such an analysis focusing on the backgrounds of informants would be better suited through another form of qualitative analysis such as grounded theory or interpretative phenomenological analysis.
Also, the sample of 25 did not include a representative of employer organisations (p.12, line 54); this is a significant omission. One might expect significant differences between categories such as professional organisation and employer. Indeed, the incorporation of employer/managerial and lay perspectives some twenty years ago was designed to bring alternative viewpoints into the decision-making	Thank you for your comment. We have already written in the limitations section of the discussion (page 14 marked copy): “Despite initial willingness to participate in the study, we struggled to recruit many non-white key informants, as well as doctors from younger age groups. Furthermore, during the recruitment process we did not collect information on some other protected characteristics such as disability, and nor did we capture the views of representatives of employer organisations. It is therefore possible that we omitted to capture some extant definitions of clinical excellence, experiences with scoring, and reflections on inequities

process.	important to some groups, organisations, and individuals.”
Finally, , while the opening section (p.4, line 53) noted wider concerns about the general lack of evidence on patient outcomes in CEA applications, this was not apparent in the findings – did interviewees not discuss it? If so, that would be significant.	Thank you for your comment. Interestingly, interviewees did not discuss the lack of evidence on patient outcomes in the assessment process. The future research section in the discussion now reads (page 14 marked copy): “Future research could involve further qualitative interviews to explore the views and experiences of assessors and applicants with the revised NCEAs scheme around the practicalities, and equitableness of the scoring process, as well as the fairness of providing evidence for the application. Further interviews could also explore whether new training materials developed by ACCEA to support prospective applicants to apply have, as intended, reduced potential barriers for marginalised groups of doctors¹². Finally, future research should explore the role of patient outcomes in the assessment of NCEA applications, given existing concerns around the general lack of evidence on patient outcomes in current assessment⁴.”
In summary, I recommend that the paper is revised and resubmitted.	Thank you for your thoughtful comments and suggestions.
Reviewer: 2 Dr. Ishanka Talagala, Ministry of Health Nutrition and Indigenous Medicine	
Reviewer comment	Response to reviewer
Congratulations to the authors on this important study. Please see the following suggestions:	Thank you for noting the importance of our study, and for your suggestions.
Page 7, line 21: “... given full information about what handling...” here, ‘what’ should be removed.	We have amended this sentence to now read (pages 7 and 15 marked copy): “Informants were given full information about handling and storage of resulting data.”
Page 14, line 6: “..anticipated to run from 2022/2023..”	We have amended this sentence to now read (page 14 marked copy): “In consultation with ACCEA, these findings are being used to inform the development of a revised scheme, anticipated to run from 2023/2024, which will aim to be more understandable and practical for assessors and applicants, and will provide clear guidance on carrying out a valid, an equitable assessment of applications.”
Need to change the years for at least 2023/2024, as we are now at the end of the year 2022, and it will take some time for the ACCEA to develop a novel system incorporating the findings of the study.	Thanks. We have updated this, as per the previous comment.